# Measurement of the Absolute Value of Cerebral Blood Volume and Optical Properties in Term Neonates Immediately after Birth Using Near-Infrared Time-Resolved Spectroscopy: A Preliminary Observation Study

**Aya Morimoto [1,2], Shinji Nakamura [1,\*], Masashiro Sugino [3], Kosuke Koyano [4], Yinmon Htun [1,2], Makoto Arioka [1], Noriko Fuke [1], Ami Mizuo [1], Takayuki Yokota [1], Ikuko Kato [1], Yukihiko Konishi [1], Sonoko Kondo [1], Takashi Iwase [1], Saneyuki Yasuda [4] and Takashi Kusaka [1]**

1. Department of Pediatrics, Faculty of Medicine, Kagawa University, 1750-1, Mikicho, Kitagun, Kagawa 761-0793, Japan; ayamoto@med.kagawa-u.ac.jp (A.M.); ymhtun0612@gmail.com (Y.H.); marioka@med.kagawa-u.ac.jp (M.A.); noriko-f@med.kagawa-u.ac.jp (N.F.); inoue26@med.kagawa-u.ac.jp (A.M.); yokotakayuki@med.kagawa-u.ac.jp (T.Y.); i-kato@med.kagawa-u.ac.jp (I.K.); lilwest@med.kagawa-u.ac.jp (Y.K.); ijichi@med.kagawa-u.ac.jp (S.K.); t.iwase@med.kagawa-u.ac.jp (T.I.); kusaka@med.kagawa-u.ac.jp (T.K.)
2. Graduate School of Medicine, Faculty of Medicine, Kagawa University, 1750-1, Mikicho, Kitagun, Kagawa 761-0793, Japan
3. Division of Neonatology, Shikoku Medical Center for Children and Adults, 2-1-1, Senyucho, Zentsuji, Kagawa 765-8507, Japan; masashiro@med.kagawa-u.ac.jp
4. Maternal Perinatal Center, Faculty of Medicine, Kagawa University, 1750-1, Mikicho, Kitagun, Kagawa 761-0793, Japan; kosuke@med.kagawa-u.ac.jp (K.K.); yas@med.kagawa-u.ac.jp (S.Y.)
\* Correspondence: shinji98@med.kagawa-u.ac.jp; Tel.: +81-87-891-2171; Fax: +81-87-891-2172

**Abstract:** The aim of this study was to use near-infrared time-resolved spectroscopy (TRS) to determine the absolute values of cerebral blood volume (CBV) and cerebral hemoglobin oxygen saturation ($ScO_2$) during the immediate transition period in term neonates and the changes in optical properties such as the differential pathlength factor (DPF) and reduced scattering coefficient ($\mu_s'$). CBV and $ScO_2$ were measured using TRS during the first 15 min after birth by vaginal delivery in term neonates who did not need resuscitation. Within 2–3 min after birth, CBV showed various changes such as increases or decreases, followed by a gradual decrease until 15 min and then stability (mean (SD) mL/100 g brain: 2 min, 3.09 (0.74); 3 min, 3.01 (0.77); 5 min, 2.69 (0.77); 10 min, 2.40 (0.61), 15 min, 2.08 (0.47)). $ScO_2$ showed a gradual increase, then kept increasing or became a stable reading. The DPF and $\mu_s'$ values (mean (SD) at 762, 800, and 836 nm) were stable during the first 15 min after birth (DPF: 4.47 (0.38), 4.41 (0.32), and 4.06 (0.28)/cm; $\mu_s'$: 6.54 (0.67), 5.82 (0.84), and 5.43 (0.95)/cm). Accordingly, we proved that TRS can stably measure cerebral hemodynamics, despite the dramatic physiological changes occurring at this time in the labor room.

**Keywords:** neonate; vaginal delivery; cerebral blood volume; cerebral hemoglobin oxygen saturation; near-infrared time-resolved spectroscopy

## 1. Introduction

Evaluation of the immediate postnatal cerebral oxygen metabolism and hemodynamics is essential for understanding extrauterine adaptation. In particular, the hemodynamic changes occurring

immediately after birth are explosive and can induce fetal distress and asphyxia. The physiological processes after birth remain unclear and include cerebral vasodilation, vasoconstriction, and oxygenation. Because the brain is the most sensitive organ system of the infant, it is important to assess its autoregulation after birth.

Near-infrared spectroscopy (NIRS) using near-infrared light (700–900 nm) enables detection of changes in the oxygenation state of hemoglobin (Hb) and water content in biological tissue. Near-infrared light is safe and penetrates deeply in the body, and NIRS has recently been applied for functional evaluation of cerebral circulation and oxygenation state in neonates [1]. Several studies have reported on the measurement of cerebral oxygenation via the tissue oxygenation index or regional saturation of oxygen ($rSO_2$) in neonates during the immediate transition after birth using NIRS [2–8]. Furthermore, NIRS can measure changes in venous Hb concentration (tHb) and cerebral blood volume (CBV), with some work showing a decrease in CBV in term neonates in the first 15 min after birth [9]. However, commonly used NIRS modalities, such as continuous wave spectroscopy, which only measures changes in the Hb concentration, and spatially resolved spectroscopy does not provide CBV but tissue oxygen saturation [1]. The absolute value of CBV is required for its use as a clinical parameter to manage circulation and determine oxygen use. Near-infrared time-resolved spectroscopy (TRS) is a unique method for calculating quantitative CBV and $ScO_2$ using a light absorption coefficient ($\mu_a$) without inducing changes in light-absorbing materials, such as Hb, because the reduced scattering coefficient ($\mu_s'$) and $\mu_a$ can be determined by resolving photon diffusion equation (PDE). Our group has already reported on the measurement of the absolute value of CBV in preterm and term neonates, determining that it sheds light on their cerebral hemodynamic development [10]. However, there have been no reports on how the absolute value of CBV is altered and whether optical properties such as the differential pathlength factor (DPF) and $\mu_s'$ are affected during this adaptation period by certain factors such as the vernix, amniotic fluid, blood, and systemic hemodynamic changes in the immediate transition period.

In this study, we used TRS to examine the absolute value of CBV and $ScO_2$ during the immediate transition period in term neonates and the changes in optical properties such as DPF and $\mu_s'$.

## 2. Materials and Methods

### 2.1. Study Design

This prospective observational study was performed at Kagawa University Hospital. The Regional Committee on Biomedical Research Ethics approved all of the included studies. Between October 2012 and April 2018, term neonates (gestational age >37 weeks, birth weight >2,300 g) were born by vaginal delivery. The study was conducted in accordance with the Declaration of Helsinki, and the protocol was approved by the local ethics committee (ethics number: H29-042). The parents of all neonates enrolled in this study provided written informed consent after receiving a full explanation of the study prior to birth. After cord clamping, routinely performed after 30 s, neonates were placed on the resuscitation table under an overhead heater. The newborn infants were dried and stimulated by using warm cotton diapers to induce effective breathing. TRS calibration was performed before each test by measuring the instrumental response function with the source and detector fibers facing each other through a neutral-density filter in a black tube [11].

Before measurement, we checked that the light intensity was adjusted within the dynamic range of the photon counter using the adult forearm. As soon as possible, another neonatologist attached the TRS probe to the newborn's right forehead (Figure 1).

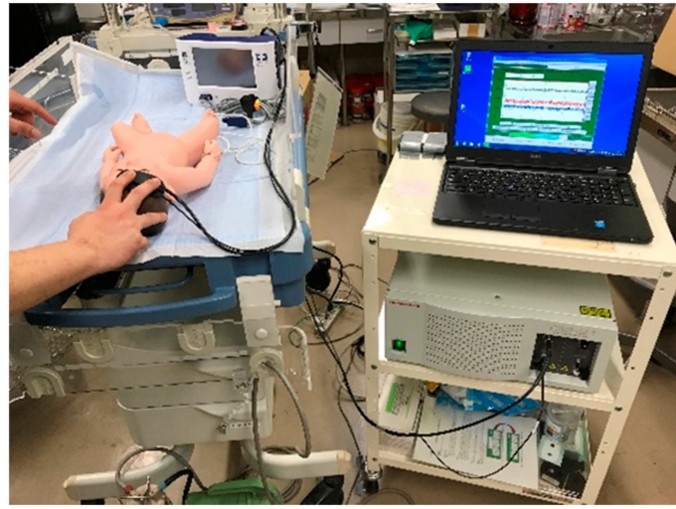

**Figure 1.** Photograph showing the actual near-infrared time-resolved spectroscopy (TRS) setting.

Then, we held the TRS transducer against the neonates' head for 15 min. Concurrently, a transcutaneous pulse oximeter was applied to the right hand (Nellcor™, COVIDIEN, Tokyo, Japan). The neonates were placed in a supine position and breathed room air. A neonatologist observed the transition of the newborn infant and recorded Apgar scores at 1 and 5 min. Resuscitation was performed following Neonatal Cardiac Pulmonary Resuscitation 2015 guidelines. We excluded the following neonates: (1) those that needed any respiratory support such as oxygen, continuous positive airway pressure, and artificial ventilation; (2) those who were hospitalized due to hypoglycemia or infection; (3) those with abnormalities such as congenital heart disease; and (4) those with an abnormal value of $\mu_a$, $\mu_s'$, and DPF retrospectively.

### 2.2. Near-Infrared Time-Resolved Spectroscopy

We used a portable three-wavelength near-infrared TRS system (TRS-21, Hamamatsu Photonics K.K., Hamamatsu, Japan). This system uses a time-correlated single-photon counting technique for detection. The system was controlled by a computer through a digital I/O interface that consisted of a three-wavelength (762, 800, and 836 nm) picosecond light pulser (PLP) as the pulse light source, a photon-counting head for single-photon detection, and signal-processing circuits for time-resolved measurement. The PLP emitted near-infrared light with a pulse duration of approximately 100 ps, a mean power of at least 200 μWat each wavelength, and pulse repetitions at a frequency of 5 MHz. The input light power to the patient was approximately 300 μW. The light from the PLP was sent to a patient from a source fiber with a length of 3 m, and the photon re-emitted from the patient was collected simultaneously by a detector fiber bundle with a length of 3 m. We obtained a set of histograms that displayed the photon flight time or re-emission profile. In this study, the emerging light was collected over a period of 1 s to exceed a photon count of at least 1000 in the peak channel of the re-emission profiles. The re-emission profiles observed at each measurement point were fitted by the time-resolved reflectance derived from the analytical solution of the PDE proposed by Patterson et al. [12,13], which is convoluted with the instrumental response function, to calculate the μa and μs' values of the head at wavelengths of 762, 800, and 836 nm.

In each iterative calculation, the analytical solution of the PDE was calculated in reflectance mode; it was then fitted to the observed re-emission profile. After determination of the $\mu_a$ and $\mu_s'$ values at three wavelengths, the oxyHb and deoxyHb concentrations were calculated from the extinction coefficients of oxyHb and deoxyHb with the following equations, based on the assumption that the background absorption was due to 85% (by volume) water.

$$\mu_{a\ 762\ nm} = \varepsilon_{762\ nm}^{oxyHb}[oxyHb] + \varepsilon_{762\ nm}^{deoxyHb}[deoxyHb] + \varepsilon_{762\ nm}^{water}[water\ volume\ fraction] \qquad (1)$$

$$\mu_{a\ 800\ nm} = \varepsilon_{800\ nm}^{oxyHb}[oxyHb] + \varepsilon_{800\ nm}^{deoxyHb}[deoxyHb] + \varepsilon_{800\ nm}^{water}[water\ volume\ fraction] \qquad (2)$$

$$\mu_{a\ 836\ nm} = \varepsilon_{836\ nm}^{oxyHb}[oxyHb] + \varepsilon_{836\ nm}^{deoxyHb}[deoxyHb] + \varepsilon_{836\ nm}^{water}[water\ volume\ fraction] \qquad (3)$$

where $\varepsilon_{\lambda\ nm}$ is the extinction coefficient at the wavelength of $\lambda$ nm and [oxyHb] and [deoxyHb] are the concentrations of oxyHb and deoxyHb, respectively. First, water absorption was subtracted from $\mu_a$ at each of the three wavelengths and then the concentrations of oxyHb and deoxyHb were estimated by applying the least-squares fitting method. The extinction coefficients for oxyHb, deoxyHb, and water shown in Table 1 were used.

**Table 1.** Extinction coefficients for oxyHb, deoxyHb, and water.

|  | oxyHb $(mM^{-1}cm^{-1})$ | deoxyHb $(mM^{-1}cm^{-1})$ | Water $(cm^{-1})$ |
|---|---|---|---|
| 762 nm | 1.4320 | 3.8145 | 0.0272 |
| 800 nm | 1.9924 | 1.9339 | 0.0204 |
| 836 nm | 2.4985 | 1.7974 | 0.0363 |

The ratio of the optical pathlength to the interoptode distance was defined as the DPF. We used the prism-type probe. The source and detector optodes were positioned on the frontal region at a 30 mm interoptode distance. The total cerebral Hb (totalHb) concentration, $ScO_2$, and CBV were calculated as follows:

$$[totalHb] = [oxyHb] + [deoxyHb] \qquad (4)$$

$$ScO_2\ (\%) = \{[oxyHb]/([oxyHb] + [deoxyHb])\} \times 100 \qquad (5)$$

$$CBV\ (ml/100\ g) = [totalHb] \times MW_{Hb} \times 10^{-6}/(tHb \times 10^{-2} \times Dt \times 10) \qquad (6)$$

where [ ] indicates Hb concentration ($\mu M$), $MW_{Hb}$ is the molecular weight of Hb (64,500), tHb is the venous Hb concentration (g/dL) and Dt is brain tissue density (1.05 g/mL).

All neonates underwent blood gas analysis, and the CBV was calculated via the venous Hb concentration at 2 h after birth. The mean values of the DPF, $\mu_a$, $\mu_s'$, CBV, and $ScO_2$ were calculated every 10 s for 15 min after birth.

## 3. Results

The study participants were seven healthy term neonates. One of the seven neonates was born via forceps delivery but the others were born via a normal vaginal delivery. Two neonates were excluded from the analysis because the $\mu_s'$ was abnormal according to previous work [10]. However, none of the neonates needed any respiratory support, had abnormalities, or required hospitalization. The gestational ages of the neonates were 38–40 weeks, and their Apgar scores at 1, 5min were 8 or more (Table 2). Five neonates (V1–V5) did not need resuscitation until 15 min after birth.

**Table 2.** Summary of neonates' clinical data in this study.

| Neonate No. | Gestational Age | Body Weight (g) | Apgar: 1 min | Apgar: 5 min | pH Umbilical Artery | Venous Hemoglobin at 2 h (g/dL) |
|---|---|---|---|---|---|---|
| V1 | 37 wk 3 d | 3,212 | 8 | 9 | 7.309 | 15.8 |
| V2 | 39 wk 6 d | 3,170 | 8 | 9 | 7.322 | 13.2 |
| V3 | 38 wk 5 d | 2,830 | 8 | 9 | 7.371 | 20.3 |
| V4 | 39 wk 4 d | 3,406 | 8 | 8 | 7.345 | 18.9 |
| V5 | 39 wk 3 d | 2,894 | 8 | 9 | 7.256 | 19.7 |

Within 2–3 min after birth, CBV showed various changes such as increases or decreases, and was then followed by a gradual decrease until 15 min, and was then stable (mean (SD) mL/100 g brain: 2 min, 3.09 (0.74); 3 min, 3.01 (0.77); 5 min, 2.69 (0.77); 10 min, 2.40 (0.61); 15 min, 2.08 (0.47)) (Figure 2).

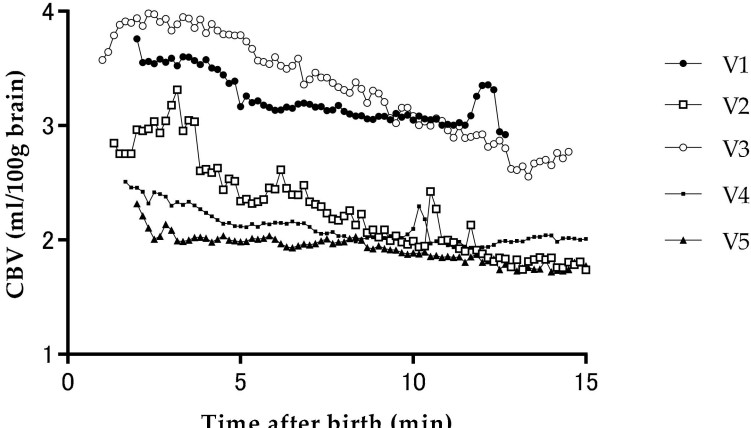

**Figure 2.** Cerebral blood volume (CBV) in five neonates during the first 15 min of life. Values are means. Compared with the reference value at 15 min, a significant decrease in CBV was observed at each time point.

$ScO_2$ showed a gradual increase, then kept increasing or became a stable reading (mean (SD)%: 2 min, 48.0 (12.3); 3 min, 53.9 (14.2); 5 min, 62.5 (13.3); 10 min, 67.2 (10.2); and 15 min, 64.3 (4.7)) (Figure 3).

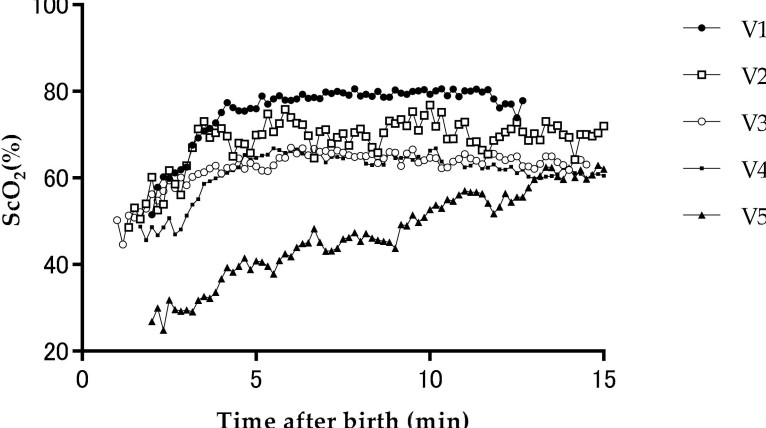

**Figure 3.** Cerebral hemoglobin oxygen saturation ($ScO_2$) in five neonates during the first 15 min of life. Values are means. $ScO_2$ showed the same pattern as arterial Hb oxygen saturation with a gradual increase, peak at 5–10 min, and then stable values.

The values of DPF, $\mu_a$, and $\mu_s'$ are shown in Table 3 and correspond to those of a previous report [10]. The optical properties were stable for the first 15 min after birth: DPF (mean (SD) at 762, 800, and 836 nm), 4.47 (0.38), 4.41 (0.32), and 4.06 (0.28)/cm; $\mu_s'$: 6.54 (0.67), 5.82 (0.84), and 5.43 (0.95)/cm.

**Table 3.** The characteristics of the mean value of DPF, μa, and μs′ in all neonates.

| Neonate No. | Time after Birth (min) | DPF | | | μa (/cm) | | | μs′ (/cm) | | |
|---|---|---|---|---|---|---|---|---|---|---|
| | | 762 nm | 800 nm | 836 nm | 762 nm | 800 nm | 836 nm | 762 nm | 800 nm | 836 nm |
| V1 | 2 | 4.06 | 4.21 | 4.08 | 0.22 | 0.17 | 0.19 | 6.42 | 5.95 | 5.97 |
| | 3 | 4.36 | 4.38 | 4.16 | 0.19 | 0.16 | 0.19 | 6.64 | 6.10 | 6.18 |
| | 5 | 4.70 | 4.47 | 4.15 | 0.15 | 0.15 | 0.17 | 6.35 | 5.89 | 5.84 |
| | 10 | 4.71 | 4.39 | 4.05 | 0.14 | 0.14 | 0.17 | 6.09 | 5.54 | 5.56 |
| | 15 | NA | NA | NA | NA | NA | NA | NA | NA | NA |
| V2 | 2 | 4.14 | 4.34 | 4.07 | 0.26 | 0.22 | 0.23 | 7.08 | 6.91 | 6.49 |
| | 3 | 4.23 | 4.46 | 4.13 | 0.26 | 0.23 | 0.26 | 7.81 | 7.57 | 7.50 |
| | 5 | 4.96 | 4.94 | 4.56 | 0.18 | 0.17 | 0.20 | 7.82 | 7.32 | 7.05 |
| | 10 | 5.21 | 5.13 | 4.63 | 0.15 | 0.15 | 0.17 | 7.36 | 6.97 | 6.59 |
| | 15 | 5.25 | 5.22 | 4.76 | 0.14 | 0.14 | 0.16 | 7.14 | 6.81 | 6.41 |
| V3 | 2 | 3.97 | 4.17 | 3.79 | 0.23 | 0.19 | 0.21 | 6.11 | 5.83 | 5.18 |
| | 3 | 4.05 | 4.14 | 3.74 | 0.22 | 0.19 | 0.21 | 6.14 | 5.70 | 5.01 |
| | 5 | 4.13 | 4.18 | 3.77 | 0.21 | 0.19 | 0.21 | 6.13 | 5.74 | 5.06 |
| | 10 | 4.44 | 4.48 | 3.99 | 0.17 | 0.16 | 0.18 | 6.04 | 5.62 | 4.97 |
| | 15 | 4.68 | 4.69 | 4.19 | 0.16 | 0.14 | 0.16 | 6.21 | 5.80 | 5.08 |
| V4 | 2 | 4.28 | 4.22 | 3.93 | 0.24 | 0.18 | 0.20 | 7.08 | 5.82 | 5.45 |
| | 3 | 4.46 | 4.35 | 4.03 | 0.22 | 0.17 | 0.19 | 7.07 | 5.87 | 5.58 |
| | 5 | 4.55 | 4.27 | 3.93 | 0.19 | 0.16 | 0.19 | 6.58 | 5.45 | 5.10 |
| | 10 | 4.81 | 4.54 | 4.12 | 0.17 | 0.16 | 0.18 | 7.02 | 5.88 | 5.42 |
| | 15 | 4.74 | 4.52 | 4.11 | 0.18 | 0.15 | 0.18 | 6.95 | 5.70 | 5.37 |
| V5 | 2 | 3.82 | 4.13 | 4.14 | 0.25 | 0.16 | 0.17 | 6.30 | 5.07 | 4.51 |
| | 3 | 4.14 | 3.95 | 3.66 | 0.22 | 0.15 | 0.15 | 5.30 | 4.35 | 3.85 |
| | 5 | 4.36 | 4.12 | 3.74 | 0.20 | 0.14 | 0.15 | 5.63 | 4.55 | 4.03 |
| | 10 | 4.36 | 4.18 | 3.76 | 0.17 | 0.14 | 0.15 | 5.63 | 4.56 | 4.03 |
| | 15 | 4.75 | 4.36 | 3.89 | 0.15 | 0.13 | 0.15 | 6.00 | 4.75 | 4.20 |

## 4. Discussion

This is the first report to use TRS to measure the absolute value of CBV and optical properties in term neonates immediately after birth. We obtained the following main results: (1) CBV shows a tendency to decrease after birth, but we found large variability in the absolute value of CBV across neonates; and (2) DPF and $\mu_s'$ were stable immediately after birth. These findings indicate that TRS can be a useful, simple, and noninvasive tool for the stable measurement of the absolute values of CBV and $ScO_2$ when we need to assess the immediate postnatal cerebral oxygen metabolism and hemodynamics in neonates.

Schwaberger et al. [14] monitored the changes in CBV during the immediate postnatal transition period after elective cesarean delivery using NIRS. The mean decrease in total Hb from 2 to 15 min after birth represented a CBV decrease of 1.0 mL/100 g brain. They hypothesized that the CBV decrease after birth was mainly caused by postnatal increases in cerebral $PaO_2$ levels. This likely reflects a physiological process involving changes in the autoregulatory capacity of cerebral vessels in reaction to increasing $pO_2$ and decreasing $pCO_2$ levels after elective cesarean delivery. Our results following natural vaginal delivery in this study are consistent with those of previous reports. We speculate that this CBV decrease can be explained in two possible ways. Firstly, in the transition period, neonates are exposed to the stress of natural labor, which would cause greater hypoxia and hypercapnia. Secondly, as this would cause cerebral vasodilation, their reactions would improve within 1–2 min after birth and CBV would decrease by 15 min after birth.

However, our findings indicate that neonates do not always show a clear decrease in CBV and that it has considerable variability. Thus, we consider that evaluation of the absolute value of CBV would more accurately reflect each characteristic cerebral hemodynamic pattern compared with the relative change in CBV alone.

Both $ScO_2$ and $SpO_2$ gradually increased until 15 min after birth, as in previous reports (Figures 3 and 4). Although there have been many reports on whether this cerebral oxyHb/totalHb ratio would become a useful parameter for monitoring oxygen during the transition period, its value remains unknown. In this study, the $ScO_2$ pattern of V3 and V4 neonates was observed to increase similarly, whereas the CBV of V3 neonates was clearly decreased and that of V4 neonates showed little decrease. These results indicated that CBV may shows a different pattern, despite the similar pattern of $ScO_2$. Therefore, it is critical to monitor both CBV

and $ScO_2$ to detect dramatic changes in cerebral hemodynamics and oxygen metabolism in the neonatal transition period. For neonates with hypoxic ischemic encephalopathy, our group reported that combined CBV plus $ScO_2$ at 24 h after birth using TRS had the best predictive ability for neurological outcomes [15]. In addition, we reported the $ScO_2$ and CBV before and after transfusion in very low birth weight infants with anemia of prematurity and suggested that CBV and $ScO_2$ may be useful markers for determining the need for transfusion [16]. Hence, monitoring of both CBV and $ScO_2$ using TRS may help to determine oxygen use in the neonatal transition period.

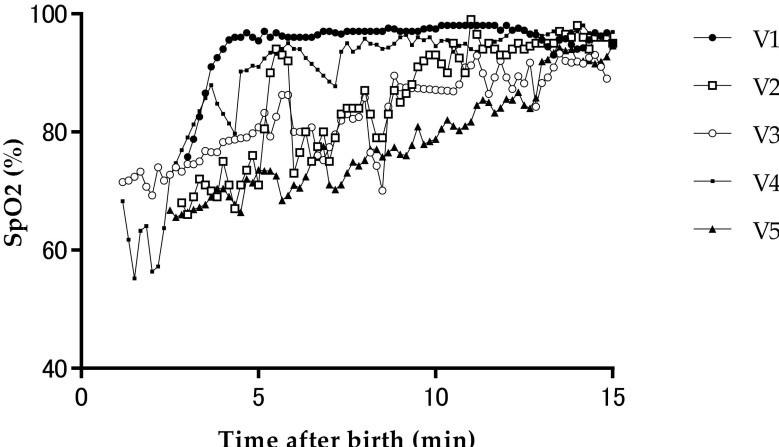

**Figure 4.** Arterial Hb oxygen saturation ($SpO_2$) in five neonates during the first 15 min of life. Values are means.

The DPF and $\mu_s'$ values found here during the immediate transition period correspond to those of the postnatal period in a previous report [10] and $\mu_s'$ value was stable from 2 to 15 min after birth. Furthermore, $\mu_s'$ is thought to depend on the conditions of tissue microstructure, such as neuron number, myelination, and edema, but is scarcely influenced by the oxygenation state and Hb concentration [10]. On the other hand, the DPF value slightly increased after birth in this study. When CBV decreased after birth, optical path length increased. This is because the contribution of photons through the deeper tissue to the temporal point spread function (TPSF) increases and the gravity of the TPSF shifts to the right due to less Hb concentration. From these results, we proved that TRS would be able to stably measure the cerebral hemodynamics, despite the dramatic physiological changes occurring in the labor room. In the next step, we will increase the size of the study population and clarify the absolute value of CBV within the standard range in term neonates, capture CBV changes in preterm neonates or neonates with asphyxia, and establish a method for resuscitation based on cerebral hemodynamics.

## 5. Limitations

First, we could not assess systemic hemodynamics and cannot provide immediate blood gas data, with data only from 2 h after birth. Second, the number of neonates in this study was too small. Thus, we will increase the number of neonates and additionally evaluate not only neonates delivered by normal vaginal delivery, but also by elective cesarean delivery, because we have to consider the possibility that the CBV might be influenced by subcutaneous congestion due to compression of the forehead during vaginal delivery. Third, our TRS system is highly portable and can measure the absolute value of CBV via the transit time of each photon through the tissue of interest. However, the system needs more than 20 min for warmup and calibration. During the immediate transition after birth, we have to adjust the attenuation level of light emission to keep the values stable. Due to neonatal body motion, another team member is required for system fitting.

## 6. Conclusions

We observed that CBV showed a tendency to decrease after birth, but had large variability in its absolute value across individual neonates, and the DPF and $\mu_s'$ values immediately after birth were stable and similar to those in previous studies. By measuring the absolute value of CBV, TRS has the potential to more accurately evaluate the cerebral hemodynamic pattern than the use of relative changes in CBV, not only $ScO_2$, during the immediate transition period, despite dramatic physiological changes occurring immediately after birth. However, the number of neonates in this study was too small to reach any definite conclusions and further work is required.

**Author Contributions:** A.M., S.N., and T.K.: substantial contributions to the conception or design of the work; Y.H, S.N., S.K., and T.K.: the necessary financial support for this project and provided study materials; A.M., S.N., M.S., K.K., M.A., T.Y., and N.F.: acquisition, analysis, or interpretation of data for the work; I.K., Y.K., S.K., S.Y., and T.I.: agreement to be accountable for all aspects of the work in ensuring that questions related to the accuracy or integrity of any part of the work are appropriately investigated and resolved.

**Funding:** This study was financially supported by JSPS KAKENHI Grant Numbers 17K10178, 16K19685, 16K10092, 17K10179, and RIKEN Healthcare and Medical Data Platform Project.

**Acknowledgments:** We thank Hamamatsu Photonics K.K. and the staff at the Faculty of Medicine Kagawa University, Kagawa, for their cooperation.

**Conflicts of Interest:** The authors have no conflicts of interest relevant to this manuscript.

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
