# Peer review of "Measurement of the Absolute Value of Cerebral Blood Volume and Optical Properties in Term Neonates Immediately after Birth Using Near-Infrared Time-Resolved Spectroscopy: A Preliminary Observation Study"

_applsci, doi:10.3390/app9102172_

Round 1
Reviewer 1 Report
Please see the attached file.

Author Response
Response to Reviewer 1 Comments
Thank you for your kind advice.
1. Line 23: “light-reduced scattering coefficient” should be changed to “reduced scattering coefficient” without“light-“.
We corrected it.
2. Line 28: “SpO2” should be spelled out as “peripheral oxygen saturation (SpO2)”.
As another reviewer pointed out this expression, we changed the sentence.
3. Lines 43-46: This sentence is too long and should be divided as the following with slight modifications indicated by underlines and double-crossed lines. “Near-infrared spectroscopy (NIRS) using near-infrared light (700–900 nm) enables detection of changes in the oxygenation state of hemoglobin (Hb) and water content in biological tissue. Near-infrared light is safe and penetrates deeply in the body well, and NIRS has recently been applied for the functional evaluation of cerebral circulation and the oxygenation state [1].
We corrected it.
4. Line 54: “estimated from an initial measurement,” is not understandable. Maybe it should be deleted.
We corrected it.
5. Line 58: “light-reduced scattering coefficient” should be changed to “reduced scattering coefficient”, and “light diffusion equation” to “photon diffusion equation (PDE).”
We corrected it.
6. Lines 78-79: “the instrumental response under the input with the receiving fibers” should be changed to “the instrumental response function with the source and detector fibers.”
We corrected it.
7. Line 85: “the TRS transducer” should be “the TRS probe.”
We corrected it.
8. Lines 108 and 110: “the photon diffusion equation” should be “the analytical solution of the PDE.”
We corrected it.
9. Line 111: “the instrumental response” should be “the instrumental response function.”
We corrected it.
10. Lines 113, 124 and caption of Table 2: “absorption coefficient” should be changed to “extinction coefficient.”
We corrected it.
11. Line 121: “In these equations, ɛλ nm is the extinction coefficient at λ nm” should be changed to “where ɛλ nm is the extinction coefficient at the wavelength of λ nm” without indenting the sentence.
We corrected it.
12. Table 2: The units for oxyHb and deoxyHb should be corrected to (mM-1cm-1) or (/(mM•cm)).
We corrected it.
13. Line 129: “The light emission and detection optodes” should be changed to “The source and detector optodes.”
We corrected it.
14. Line 149: This line should be removed.
We removed the subtitle “CBV and ScO2”.
15. Table 3: “N1” to “N5” for “Neonate no.” should be changed to “V1” to “V5”.
We corrected it.
16. Line 197: “looked similarly increased” might be changed to “was observed to increase similarly”.
We corrected it.
17. Lines 208-209: It is described that the DPF and μs′ values … were stable …” However, from Table 3 it looks that the DPF slightly increases with time after birth while μs′ values were relatively unchanged. How do the authors argue about it?
As you pointed out, the DPF slightly increased after birth.
We consider as follows;
When CBV decreased after birth, optical path length got increased. Because the near-infrared light can travel through tissue more far due to less Hb concentration. On the other hand, tissue structure did not change, therefore µ’s did not change.
Reference;
C. Dean Kurth, at al. Cerebral Hemoglobin and Optical Pathlength Influence Near-Infrared Spectroscopy Measurement of Cerebral Oxygen Saturation. Anesthesia & Analgesia. 84(6):1297-1305, 1997
We added this sentences and citation to line 211-213. Please correct it.

Reviewer 2 Report
This work demonstrated that near-infrared time-resolved spectroscopy (NITRS) can be used to measure absolute values of cerebral blood volume and cerebral hemoglobin oxygen saturation in neonates while the optical properties remain comparable. Despite the small sample/subject size, the authors validated that NITRS would be an effective technique for future applications. There are only few comments that need to be addressed.
1. Lines 25-26 “CBV peaked about 1 to 2 mins…”: The statement is not necessarily true because high variability was observed across different subjects as mentioned by the authors (lines 174-175) and a peak did not always exist (Fig. 2). The authors may want to revise the statement throughout the manuscript.
2. Lines 27-28 “ScO2 showed the same pattern…”: Similar to the comment #1, a peak did not exist in all subjects and some of them increased over time without any peaks; for example in SpO2, v2 and v5.
3. Lines 50-54 “However, commonly used NIRS…”: Need some citations.
4. Lines 142-143: Please specify what abnormal light-reduced scattering coefficients meant when excluding the two neonates.
5. Page 4: Labels for Table 1 and 2 should be switched.
6. Figs. 2, 3 and 4: Please report the acquisition rate used in TRS. It looks like each curve in Figs. 2, 3 and 4 had different starting/initiating point in terms of time (some curves started at 1 min while the others started at 2-3 min) and might be applied with different acquisition rate.
Author Response
Answer to Reviewer2
Thank you for your advice.
1. Lines 25-26 “CBV peaked about 1 to 2 mins…”: The statement is not necessarily true because high variability was observed across different subjects as mentioned by the authors (lines 174-175) and a peak did not always exist (Fig. 2). The authors may want to revise the statement throughout the manuscript.
Your suggestion is correct. We fixed Lines 25-26 sentence as follows: CBV tended to peak or maintain around few mins, then decreased gradually until 15min.
2. Lines 27-28 “ScO2 showed the same pattern…”: Similar to the comment #1, a peak did not exist in all subjects and some of them increased over time without any peaks; for example in SpO2, v2 and v5.
Your suggestion is also correct as #1. We fixed Line 27-28 sentence as follows: ScO2 showed a gradual increase, then kept increasing or became stable readings.
3. Lines 50-54 “However, commonly used NIRS…”: Need some citations.
We added a citations [1]. Please see the reference list.
4. Lines 142-143: Please specify what abnormal light-reduced scattering coefficients meant when excluding the two neonates.
Previously, our group reported about the neonates’s values of light reduced scattering coefficient measured by TRS, and revealed that their range was 4-8/cm. In this study, two neonates’ values were out of range, therefore we decided to exclude them. However, it is unclear the reason why their values were out of range.
5. Page 4: Labels for Table 1 and 2 should be switched.
We’re sorry for the careless mistake. We fixed it.
6. Figs. 2, 3 and 4: Please report the acquisition rate used in TRS. It looks like each curve in Figs. 2, 3 and 4 had different starting/initiating point in terms of time (some curves started at 1 min while the others started at 2-3 min) and might be applied with different acquisition rate.
The acquisition rate was 1sec in TRS.

Reviewer 3 Report
This is a good study to use TRS to measure the absolute value in after birth neonates. They can show the consistent results in all five subjects. I have a few comments to improve the manuscript:
(i) Please explain what is the objective to measure after 15-min? Is that because the IRB approval or any scientific reasons behind that time?
(ii) Please explain by equation and citation when they convert the raw data become oxy- and deoxy-hemoglobin (for example in CW-fNIRS use modified Beer-Lambert law and FD-fNIRS use transport Boltzmann equation.
(iii) Please give the numbers in every equation.
(iv) Please explain how to calculate DPF as shown in Table 3.
(v) The numbers in x- and y-label are not clear in Figs. 2-4. Please change it.
Author Response
Answer for reviewer 3
Thank you for your advice.
(i) Please explain what is the objective to measure after 15-min? Is that because the IRB approval or any scientific reasons behind that time?
We have two reasons to measure cerebral hemodynamic changes after 15 min.
At first, cerebral hemodynamics changed dramatically within 15 min after birth as many previous studies already revealed, therefore we focused on this periods to measure CBV and ScO2 using TRS in this study.
Secondly, this immediate transition 15min period after birth is critical to resuscitate neonates needed respiratory or circulation support after birth. We consider that monitoring CBV and ScO2 may be helpful parameter to resuscitate this transition period in the future.
(ii) Please explain by equation and citation when they convert the raw data become oxy- and deoxy-hemoglobin (for example in CW-fNIRS use modified Beer-Lambert law and FD-fNIRS use transport Boltzmann equation.
We mentioned at method as follows;
The re-emission profiles observed at each measurement point were fitted by the photon diffusion equation proposed by Patterson’s principle in order to calculate the values of absorption
coefficient (µa) and reduced scattering coefficient (µs’) of the head at wavelengths of 762, 800, and 836 nm.
Reference;
[12] Patterson, M.S.; Chance, B.; Wilson, B.C. Time resolved reflectance and transmittance for the non-invasive measurement of tissue optical properties. Appl Opt 1989, 28, 2331-2336, doi:10.1364/ao.28.002331.
[13] Ijichi, S.; Kusaka, T.; Isobe, K.; Islam, F.; Okubo, K.; Okada, H.; Namba, M.; Kawada, K.; Imai, T.; Itoh, S. Quantification of cerebral hemoglobin as a function of oxygenation using near-infrared time-resolved spectroscopy in a piglet model of hypoxia. Journal of biomedical optics 2005, 10, 024026, doi:10.1117/1.1899184.
(iii) Please give the numbers in every equation.
We gave the numbers in equations.
(iv) Please explain how to calculate DPF as shown in Table 3.
We calculated “Pathlength divided by 3(cm)* equal DPF”.
*3 cm: interoptode distance
(v) The numbers in x- and y-label are not clear in Figs. 2-4. Please change it.
We fixed the boldfaced type label.
